# (Nearly) Efficient Algorithms for the Graph Matching Problem on Correlated Random Graphs

**Boaz Barak**[*]
School of Engineering and Applied Science
Harvard University
Cambridge, MA, 02138
b@boazbarak.org

**Chi-Ning Chou**[*]
School of Engineering and Applied Science
Harvard University
Cambridge, MA, 02138
chiningchou@g.harvard.edu

**Zhixian Lei**[*]
School of Engineering and Applied Science
Harvard University
Cambridge, MA, 02138
leizhixian.research@gmail.com

**Tselil Schramm**[*]
School of Engineering and Applied Science
Harvard University
Cambridge, MA, 02138
tselil@seas.harvard.edu

**Yueqi Sheng**[*]
School of Engineering and Applied Science
Harvard University
Cambridge, MA, 02138
ysheng@g.harvard.edu.

## Abstract

We consider the *graph matching/similarity* problem of determining how similar two given graphs $G_0, G_1$ are and recovering the permutation $\pi$ on the vertices of $G_1$ that minimizes the symmetric difference between the edges of $G_0$ and $\pi(G_1)$. Graph matching/similarity has applications for pattern matching, computer vision, social network anonymization, malware analysis, and more. We give the first efficient algorithms proven to succeed in the correlated Erdös-Rényi model (Pedarsani and Grossglauser, 2011). Specifically, we give a polynomial time algorithm for the *graph similarity/hypothesis testing* task which works for every constant level of correlation between the two graphs that can be arbitrarily close to zero. We also give a quasi-polynomial ($n^{O(\log n)}$ time) algorithm for the *graph matching* task of recovering the permutation minimizing the symmetric difference in this model. This is the first algorithm to do so without requiring as additional input a "seed" of the values of the ground truth permutation on at least $n^{\Omega(1)}$ vertices. Our algorithms follow a general framework of counting the occurrences of subgraphs from a particular family of graphs allowing for tradeoffs between efficiency and accuracy.

## 1 Introduction

The *graph matching* and *graph similarity* problems are well-studied computational problems with applications in a great many areas. Some examples include machine learning [1], computer vision [2], pattern recognition [3], computational biology [4, 5], social network analysis [6], de-

---

[*]Supported by NSF awards CCF 1565264 and CNS 1618026.

anonymization [7], and malware detection [8].[2] The *graph matching* problem is the task of computing, given a pair $(G_0, G_1)$ of $n$ vertex graphs, the permutation

$$\pi^* = \arg \min_{\pi \in \mathcal{S}_n} \|G_0 - \pi(G_1)\|_0 \tag{1}$$

where we identify the graphs with their adjacency matrices, and write $\pi(G_1)$ for the matrix obtained by permuting the rows and columns according to $\pi$ (i.e., the matrix $P^\top G_1 P$ where $P$ is the permutation matrix corresponding to $\pi$). The *graph similarity* problem is to merely determine whether or not $G_0$ is similar to $G_1$ or more generally to obtain an efficiently computable *distance measure* on $G_0$ and $G_1$ that provides a rough approximation to $\min_{\pi \in \mathcal{S}_n} \|G_0 - \pi(G_1)\|_0$.

In this paper we obtain new algorithms with provable guarantees for both problems. These problems are NP hard in the worst case[3] and hence our focus is on average case complexity and specifically the *correlated Erdös-Rényi model* introduced by [11] and studied in [6, 12, 13, 14, 15, 16]. For $n$ a positive integer and $0 < p, \gamma < 1$, the *correlated Erdös-Rényi model with parameters* $n, p, \gamma$ is the following distribution over triples $(G_0, G_1, \pi)$ where $G_0, G_1$ are $n$-vertex graphs and $\pi$ is permutation on $[n]$: (i) We sample a base graph $B$ from the Erdös-Rényi random graph distribution $\mathbb{G}(n, p)$, (ii) We let $G, G'$ to be two independent random subgraphs of $B$, where every edge from $B$ is kept in $G$ and $G'$ with probability $\gamma$ independently, (iii) We choose a random permutation $\pi$ and output $(G, \pi(G'), \pi)$.[4] We denote this distribution by $\mathcal{D}_{\text{struct}}(n, p; \gamma)$. We say that $(G_0, G_1)$ are sampled from $\mathcal{D}_{\text{struct}}(n, p; \gamma)$ if they are obtained by sampling $(G_0, G_1, \pi)$ from this distribution and discarding the permutation $\pi$. We use $\mathcal{D}_{\text{null}}(n, p; \gamma)$ for the product distribution $\mathbb{G}(n, p\gamma) \times \mathbb{G}(n, p\gamma)$. Note that the marginals over $G_0, G_1$ are the same in both $\mathcal{D}_{\text{struct}}$ and $\mathcal{D}_{\text{null}}$ but the two graphs are correlated in the former distribution and independent in the latter. We consider the following two computational problems:

**Graph similarity: hypothesis testing.** Given $(G_0, G_1)$ sampled *either* from $\mathcal{D}_{\text{struct}}(n, p; \gamma)$ *or* from $\mathcal{D}_{\text{null}}(n, p; \gamma)$. The goal is to distinguish which distribution the input $(G_0, G_1)$ was sampled from. Graph similarity (for general models) has been proposed as a tool for malware detection [17, 18], chemical structure similarity [19, 20], comparing biological networks [21] and more.

**Graph matching: recovery.** Given $(G_0, G_1)$ sampled from $\mathcal{D}_{\text{struct}}(n, p; \gamma)$, the goal is to recover the permutation $\pi$. Graph matching has a long history in pattern recognition [3], social network de-anonymization [7] and more.

## 1.1 Our contributions

It is known as long as $p\gamma^2 \gg \log n/n$, if $(G_0, G_1, \pi)$ is drawn from $\mathcal{D}_{\text{struct}}(n, p\gamma)$ then $\pi$ will be the minimizer of the right-hand side of (1), but prior to this work it was not known whether there is an efficient algorithm to recover $\pi$ (see Section 1.2 for related work). In this work we give algorithms for both the hypothesis testing and recovery problems on the correlated Erdös-Rényi model $\mathbb{G}(n, p; \gamma)$ for every constant (and even slightly sub-constant) $\gamma$ and a wide range of $p$.

**Theorem 1.1** (Hypothesis testing). *For every $\epsilon > 0$, sufficiently small $\delta > 0$, and $\gamma > 0$ there is a polynomial time algorithm A that distinguishes with success probability at least $1 - \epsilon$ between the case that $(G_0, G_1)$ are sampled from $\mathcal{D}_{struct}(n, n^{\delta-1}; \gamma)$ and the case that they are sampled from $\mathcal{D}_{null}(n, n^{\delta-1}; \gamma)$.*

**Theorem 1.2** (Recovery). *For every $\epsilon > 0$, sufficiently small $\delta > 0$, and $\gamma > 0$, there is a randomized algorithm A with running time $n^{O(\log n)}$ such that with probability at least $1 - \epsilon$ over $(G_0, G_1, \pi^*) \sim \mathcal{D}_{struct}(n, n^{\delta-1}; \gamma)$ and over the choices of A, we have $A(G_0, G_1) = \pi^*$.*

These are the first algorithms that run in better than exponential time for these problems (see Table 1). While the main contribution of this paper is theoretical, we believe that our techniques are of independent interest and applicability beyond the correlated Erdös-Rényi model. Key to our work is the notion of identifying a large family of subgraphs (a "flock of black swans") each of which is

highly unlikely to occur as a subgraph in a random graph but satisfying some near-independence conditions that imply that with high probability *some* members of the family will occur. The existence of such a family is by no means easy to establish— showing this accounts for much of the technical work in this paper and there are still ranges of parameters for which we conjecture that such families exist but have not been able to prove so. However, for any given distribution of graphs, one can search for subgraphs that will serve as useful features for both graph similarity and graph matching.

| Paper | Algorithm | Runtime |
|---|---|---|
| Cullina & Kivayash [15, 16] | exhaustive search (information theoretic bound) | $O(n!)$ |
| Yartseva & Grossglauser [12] | percolation from seed set | $\exp(n^{1-\delta-\Theta(\delta^2)})$ |
| **This paper** | subgraph matching | $n^{O(1)}$ for testing $n^{O(\log n)}$ for recovery. |

Table 1: Comparison with prior algorithms rigorously analyzed for recovery or testing in the correlated Erdös-Rényi model, when $(G_0, G_1, \pi) \sim \mathcal{D}_{\text{struct}}(n, n^{\delta-1}; \gamma)$ for $\delta > 0$. Prior algorithms were analyzed in this model for the recovery task which subsumes testing. See related work section for a full discussion.

**Remark 1.3.** While we state our results for "sufficiently small" $\delta$ they actually hold in a broader setting (i.e., for $0 < \delta \le 1/153$ or $\frac{2}{3} \le \delta < 1$). Under a certain combinatorial conjecture our algorithms works for all $0 < \delta < 1$, see supplementary material.

## 1.2 Related work

There has been significant amount of work on the correlated Erdös-Rényi model. Cullina and Kivayash [15, 16] precisely characterized the parameters $p, \gamma$ for which information theoretic recovery is possible. Specifically, they showed recovery is possible (in the information-theoretic sense, via an exhaustive search over all permutations) if $p\gamma^2 > \frac{\log n + \omega(1)}{n}$ and impossible when $p\gamma^2 < \frac{\log n - \omega(1)}{n}$. Yartseva and Grossglauser [12] analyzed a simple algorithm known as *Percolation Graph Matching (PGM)*, which was used successfully by Narayanan and Shmatikov [7] to de-anonymize many real-world networks. (Similar algorithms were also analyzed by [6, 14, 13].) This algorithm starts with a "seed set" $S$ of vertices in $G_0$ that are mapped by $\pi$ to $G_1$, and for which the mapping $\pi|_S$ is given. It propagates this information according to a simple percolation, until it recovers the original permutation. Yartseva and Grossglauser gave precise characterization of the size of the seed set required as a function of $p$ and $\gamma$ [12]. Specifically, in the case that $\gamma = \Omega(1)$ and $p = n^{-1+\delta}$ (where the expected degree of $G_0$ and $G_1$ is $\Theta(n^\delta)$), the size of the seed set required is $|S| = n^{1-\delta-\Theta(\delta^2)}$. In the general setting when one is *not* given such a seed set, we would require about $n^{|S|}$ steps to obtain it by brute force, which yields an $\exp(n^{\Omega(1)})$ time algorithm in this regime. Lyzinski et al. [22] also gave *negative results* for popular convex relaxations for graph matching on random correlated graphs.

We use a variant on the PGM algorithm as a component in our work to "boost" an initial partial permutation into a the full knowledge. As part of that, we extend the analysis of PGM to show it works even in the case where the partial assignment is noisy and the seed set itself might not be random but rather can be adversarially chosen, see Lemma 4.2.

There have been many works on heuristics for both graph matching and graph similarity (see the surveys [9, 10]). In particular [23, 24, 21, 25, 26] studied the graph similarity problem of deciding whether two graphs are similar to one another. [27, 28, 29, 30] trained a deep neural network to extract features of graphs for graph similarity.

## 2 Approaches and Techniques

In this section, we illustrate our approach and techniques. For simplicity and concreteness, we set the noise parameter $\gamma$ to half, and focus on the hypothesis testing task of distinguishing whether graphs $(G_0, G_1)$ are sampled from $\mathcal{D}_{\text{null}}(n, n^{\delta-1}; \frac{1}{2})$ or $\mathcal{D}_{\text{struct}}(n, n^{\delta-1}; \frac{1}{2})$ for some small constant $\delta > 0$.

**Warm-up: degree sequence.** Since graph matching is a noisy version of graph isomorphism, as a warm-up let us consider one of the most common heuristics for graph isomorphism which measures similarity of the graphs using their *degree sequence*. Namely, using the vector of sorted degrees of the

vertices in the graph as a feature vector. If $G_0$ and $G_1$ were isomorphic then the two vectors will be identical, while for two independent graphs the vectors are highly likely to differ. While this heuristic is quite successful in the setting of (noiseless) graph isomorphism setting in getting at least an initial assignment, it completely fails in our noisy setting of the graph matching and similarity problems. Intuitively, this is due to the fact degrees in a random graph are highly concentrated (generally of the form $pn \pm O(\sqrt{pn})$) and so even adding a small constant amount of noise will have a large effect on the order of the vertices in the sorting, hence making corresponding coordinates of the two vectors independent from one another. A similar phenomenon holds for the case where we use the sorted top eigenvectors of the adjacency matrix as a feature vector. While the degree and eigenvectors are poorly suited for handling noisy graphs, it turns out we can design better features by looking at *subgraph counts* for carefully chosen families of graphs. This is what we do.

## 2.1 The "black swan" approach

Our approach can be viewed as *"using a flock of black swans"*. Specifically, for each $b \in \{0, 1\}$, we map the graphs $G_0, G_1$ into a pair of feature vectors $v^0, v^1 \in \mathbb{Z}^k$ as follows: Let $\mathcal{H} = \{H_1, \ldots, H_k\}$ be a carefully chosen family of small graphs. Next, for $b \in \{0, 1\}$ and $j \in \{1, 2, \ldots, k\}$, define the $j^{\text{th}}$ coordinate of $v^b$ to be the number of occurrences of the graph $H_j$ as being a subgraph of $G_b$.[5] We choose the family $\mathcal{H}$ to satisfy the following two conditions:

**"Black swan":** For every $H \in \mathcal{H}$, the probability that $H$ occurs as a subgraph of a random graph $G$ from $\mathbb{G}(n, p)$ is a small number $\mu \ll 1$.

**Pairwise independence (informal):** For $H \neq H'$ in $\mathcal{H}$, the probability both $H$ and $H'$ both occur as subgraphs in a random graph $G$ from $\mathbb{G}(n, p)$ is up to a constant factor the product of the probabilities that each one of them occurs individually.

Before going to the details of the technical properties of black swans, let us first take a look at why this would be useful for the hypothesis testing problem. Let's assume for simplicity that all the graphs in $\mathcal{H}$ have $e$ edges for some constant $e$. If $G_0, G_1$ are $\gamma$ correlated then for every $j \in \{1, 2, \ldots, k\}$, the coordinates $v_j^0$ and $v_j^1$ will have a correlation of $\gamma^{2e}$. In contrast, if $G_0, G_1$ are independently chosen then $v^0$ and $v^1$ are completely independent and hence $v_j^0$ and $v_j^1$ have zero correlation. The number $\gamma^e$ is very small, but the pairwise independence condition implies that if the size $|\mathcal{H}|$ of the family is much larger than $(1/\gamma)^{2e}$ then the vectors $v^0$ and $v^1$ will have a significantly larger inner product in the correlated case than they do in the null case. We instantiate the above idea into a hypothesis testing algorithm in Section 3.

**Remark 2.1** (Black-swan based algorithm for recovery). The above approach can be extended to the *recovery* problem as well. The idea is that for every vertex $i$ of $G_b$ we define a vector $v^{b,i} \in \mathbb{Z}^k$ such that for all $\ell \in [k]$, $v_\ell^{b,i}$ is equal to the number of subgraphs of $G_b$ isomorphic to $H_\ell$ that *touch the vertex $i$*. The intuition is that for vertices $i$ of $G_0$ and $j$ of $G_1$, the vectors $v^{0,i}$ and $v^{1,j}$ are much more likely to have significant inner product if $\pi(i) = j$, this can be used to obtain partial information on the permutation that can later be "boosted" to recover the full permutation. We instantiate the above idea into a recovery algorithm in Section 4.

## 2.2 Constructing the black swan family

We now describe more precisely the properties that our family $\mathcal{H}$ of "black swans" or *test graphs* needs to satisfy so the above algorithm will succeed. It is encapsulated in the following theorem:[6]

**Theorem 2.2** (General overview of test graph properties). *For any rational scalar $d \in (2, 2 + \frac{1}{76})$ or $d \in \mathbb{Z}_{\geq 3}$ or $d \geq 6$, and integer $v_0$ there exists $v \geq v_0$ and set $\mathcal{H}_d^v$ of $v$-vertex graphs s.t.:*

1. *(Low likelihood of appearing) Every $H \in \mathcal{H}_d^v$ has average degree $d$. That is, the number of edges of $H$ is $e = dv/2$.*

2. (*Strong strict balance*) *For every* $H \in \mathcal{H}_d^v$ *and induced subgraph* $H'$ *of* $H$ *with* $e(1 - \epsilon)$ *edges and* $v'$ *vertices satisfies* $e'/v' < e/v - \eta$ *for a constant* $\eta$ *depending only on* $\epsilon$ *and* $d$ .[7]

3. *Every* $H \in \mathcal{H}_d^v$ *has no non-trivial automorphisms.*

4. (*Pairwise near independence*) *For every pair of distinct graphs* $H, H' \in \mathcal{H}_d^v$ *if* $J$ *is a shared subgraph of* $H$ *and* $H'$ *of* $e''$ *edges and* $v''$ *vertices then* $e''/v'' < e/v - \eta'$ *where* $\eta'$ *is a constant depending only on* $d$.

5. (*Largeness*) *The size of the family is* $|\mathcal{H}| = v^{cv}$ *where* $c$ *is a constant depending only on* $d$.

The proof of Theorem 2.2 is quite involved, and we leave it to the supplementary materials. Here, we sketch the construction of $\mathcal{H}_d^v$ where $d = 2 + \delta$ for a small constant $\delta > 0$. This is the most interesting parameter regime, as it corresponds to the sparse graph case where the degree of $G_0, G_1$ is $\sim n^\delta$. We can express the number $(1 - \delta)/1.5\delta$ as a convex combination $k\alpha + (k + 1)(1 - \alpha)$ of two integers $k, k + 1$. We choose a large enough integer $v$ so that $\delta v$, $1.5\delta v$ and $\alpha 1.5\delta v$ are all integers. Now we choose a random three-regular graph $H$ on $\delta v$ vertices (and hence $1.5\delta v$ edges), pick $1.5\alpha\delta v$ of the edges of $H$ uniformly at random, and replace them with paths of length $k$ (i.e., subdivide the edge with $k$ vertices) and replace the remaining $(1 - \alpha) \cdot 1.5\delta v$ edges with $k + 1$ length paths. The resulting graph $H'$ will have average degree $2 + \delta$ as desired. The bulk of the analysis is to prove that with high probability the graph $H'$ will satisfy the strong strict balance property, and moreover we can repeat this process $v^{\Omega(v)}$ times and get a family of graphs, every pair of which satisfies the pairwise near independence property. See Figure 1 for an example.

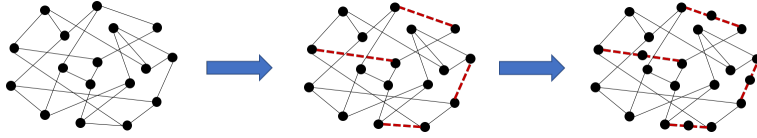

Figure 1: An example of the construction where $d = 2 + \delta$ and $k = 0$.

# 3 Algorithm for Hypothesis Testing

In this section, we describe the algorithm for the hypothesis testing based on the "black swan" approach introduced in Section 2. Let $\mathcal{H}$ be the family of graphs constructed in Theorem 2.2, we define the following *correlation polynomial*.

$$P_\mathcal{H}(G_0, G_1) = \frac{1}{|\mathcal{H}|} \sum_{H \in \mathcal{H}} \left( X_H(G_0) - \mathbb{E}_{G \sim \mathbb{G}(n, p\gamma)} X_H(G) \right) \left( X_H(G_1) - \mathbb{E}_{G \sim \mathbb{G}(n, p\gamma)} X_H(G) \right) .$$

Intuitively, the expectation of $P_\mathcal{H}(G_0, G_1)$ is zero under $\mathcal{D}_{\text{null}}$ but large under $\mathcal{D}_{\text{struct}}$. Specifically, we prove the following theorem:

**Theorem 3.1.** *For any* $n$ *large enough, sufficiently small* $\delta > 0$, *and any* $\gamma \in (0, 1)$, *let* $\mathcal{H} = \mathcal{H}_d^v$ *obtained from Theorem 2.2 where* $d = \frac{2}{1-\delta}$ *and* $|\mathcal{H}_d^v| \geq (400/\gamma^2)^{dv}$. *Then* $\mathbb{E}_{\mathcal{D}_{\text{null}}}[P_\mathcal{H}(G_0, G_1)] = 0$, *and* $\mathbb{E}_{\mathcal{D}_{\text{struct}}}[P_\mathcal{H}(G_0, G_1)] \geq 40 \cdot \max \left( \text{Var}_{\mathcal{D}_{\text{null}}} \left( P_\mathcal{H}(G_0, G_1) \right)^{1/2}, \text{Var}_{\mathcal{D}_{\text{struct}}} \left( P_\mathcal{H}(G_0, G_1) \right)^{1/2} \right)$ *where all distributions above are for* $n$ *vertices,* $p = n^{\delta - 1}$ *and noise* $\gamma$.

The proof of Theorem 3.1 is provided in the supplementary materials. The degree of the polynomial $P_\mathcal{H}(G_0, G_1)$ is $2e$ where $e$ is the number of edges in any member of the family, and so its number of monomials (and hence computation time) will be $n^{O(e)} = n^{O(1)}$ where the constant in the $O(1)$ depends on the size of the representation of $(1 - \delta)^{-1}$ as a ratio of two integers. Combining Theorem 3.1 with Chebyshev's inequality, the following algorithm solves the hypothesis testing problem in the parameter regime stated in Theorem 1.1.

**Algorithm 1** HYPOTHESISTESTING

---

**Input:** Parameters $n, p, \gamma$ where $p = n^{\delta-1}$. Graphs $G_0, G_1$ sampled from either $\mathcal{D}_{\text{null}}(n, p; \gamma)$ or $\mathcal{D}_{\text{struct}}(n, p; \gamma)$.
**Output:** "$(G_0, G_1)$ came from $\mathcal{D}_{\text{null}}$" or "$(G_0, G_1)$ came from $\mathcal{D}_{\text{struct}}$".
  1: $d \leftarrow \frac{2}{1-\delta}$.
  2: Choose $v$ be a sufficiently large even number such that $v^c > 400/\gamma^2$ where $c$ is the constant from Theorem 2.2 so that $|\mathcal{H}_d^v| \geq v^{\frac{cd}{2}v}$.
  3: $\mathcal{H} \leftarrow \mathcal{H}_d^v$ where $\mathcal{H}_d^v$ is obtained from Theorem 2.2.
  4: Compute $\mu_{\text{struct}} \leftarrow \mathbb{E}_{(G_0', G_1') \sim \mathcal{D}_{\text{struct}}(n, p; \gamma)}[P_{\mathcal{H}}(G_0', G_1')]$.
  5: **if** $P_{\mathcal{H}}(G_0, G_1) > \frac{1}{3}\mu_{\text{struct}}$ **then** Output "$(G_0, G_1)$ came from $\mathcal{D}_{\text{struct}}$".
  6: **else** Output "$(G_0, G_1)$ came from $\mathcal{D}_{\text{null}}$". **end if**

---

## 4 Algorithm for Recovery

In this section we present our algorithm for the recovery (i.e., graph matching) task. All proofs are provided in the supplemantary material. Our algorithm follows the following general template:

---

**Algorithm 2** RECOVERY

---

**Input:** Parameters $n, p, \gamma$ and graphs $G_0, G_1$ sampled from $\mathcal{D}_{\text{struct}}(n, p; \gamma)$.
**Output:** A permutation $\pi \in \mathcal{S}_n$.
  1: $\mathcal{H} \leftarrow$ INITIALIZERECOVERY$(n, p, \gamma)$.     ▷ Initialize a graph family $\mathcal{H} = \mathcal{H}_{d'}^v$ by Theorem 2.2.
  2: $\pi_0 \leftarrow$ PARTIALASSIGNMENT$(n, p, \gamma, G_0, G_1, \mathcal{H})$.     ▷ Find an initial partial assignment $\pi_0$.
  3: $\pi \leftarrow$ BOOSTING$(n, p, \gamma, G_0, G_1, \pi_0)$.  ▷ Boost the partial assignment $\pi_0$ to final assignment $\pi$.
  4: **return** $\pi$.

---

There are three steps in the above general template algorithm RECOVERY, each of them is of independent interest. In the first step, one construct a family of subgraphs of nice structure so that in the second step these subgraphs can be used to efficiently come up with a *partial assignment $\pi_0$* to the recovery problem. A partial assignment correctly matches a good fraction of vertices between $G_0$ and $G_1$, however, one does not know which vertices are correctly matched. Thus, in the last step, the boosting algorithm transforms an arbitrary partial assignment $\pi_0$ to a final assignment $\pi$ that correctly matches every vertex. The main contribution of this paper lies in the first two steps which use the black swan approach while the last step is a variant of the previous seed-set based algorithms. In the following, we instantiate RECOVERY using the test graph family constructed in Theorem 2.2 and prove Theorem 1.2.

**Step 1: Construct graph family**   Here we describe the algorithm INITIALIZERECOVERY as follows. For $p = n^{\delta-1}$, if $0 < \delta < \frac{1}{153}$, choose $v = \Theta(\log n)$, to be the smallest even integer so that $\lambda v$ is also an integer, for some $\lambda \in \left(\frac{2\delta}{1-\delta}, \frac{2\delta}{1-\delta} + \frac{\log\log n}{\log n}\right)$ and set $d' = 2 + \lambda$. If $\frac{2}{3} \leq \delta < 1$, choose $v = \Theta(\log n)$, to be the smallest even integer so that there is some $d' \in \left(\frac{2\delta}{1-\delta}, \frac{2\delta}{1-\delta} + \frac{\log\log n}{4\log n}\right)$, so that $(d' - \lfloor d' \rfloor v)$ is also an integer. Finally, pick $\mathcal{H}$ to be $\mathcal{H}_{d'}^v$ where $\mathcal{H}_{d'}^v$ is obtained from Theorem 2.2.

**Step 2: Partial assignment**   The second part of the recovery algorithm is a procedure in finding a *noisy seed set*. Specifically, if $(G_0, G_1, \pi^*)$ are sampled from $\mathcal{D}_{\text{struct}}(n, p; \gamma)$, and $0 < \theta, \eta \leq 1$ are some constants then an $(\theta, \eta)$ *partial assignment* is a partial function $\pi : V(G_0) \to V(G_1)$ that is one-to-one defined on at least $\theta$ fraction of the inputs s..t. for at least $\eta$ fraction of the inputs $u$ on which $\pi$ is defined, $\pi(u) = \pi^*(u)$. We prove that algorithm PARTIALASSIGNMENT below gives a $\left(\frac{n}{O(\log\log n)}, 1 - o(1)\right)$-partial assignment with probability $1 - o(1)$ over the Erdös-Rényi model and the randomness of the algorithm.

**Lemma 4.1.** *Suppose that $(G_0, G_1) \sim \mathcal{D}_{\text{struct}}(n, p; \gamma)$ and $\mathcal{H} = \mathcal{H}_d^v$ from INITIALIZERECOVERY. Then under the conditions of Theorem 1.2, PARTIALASSIGNMENT outputs a $\left(\frac{n}{\log v}, 1 - \frac{1}{v^{1/8}}\right)$-partial assignment with probability $1 - o(1)$ over the choice of $(G_0, G_1) \sim \mathcal{D}_{\text{struct}}(n, p; \gamma)$ and the randomness of the algorithm.*

**Algorithm 3** PARTIALASSIGNMENT

**Input:** Parameters $n, p, \gamma$, graphs $G_0, G_1$ sampled from $\mathcal{D}_{\text{struct}}(n, p; \gamma)$, and a family of graphs $\mathcal{H}$.
**Output:** A permutation $\pi_0 \in \mathcal{S}_n$.
1: $v \leftarrow |V(H)|, \ e \leftarrow |E(H)|, \ \forall H \in \mathcal{H}$.
2: $d' \leftarrow \frac{2e}{v}$.
3: $\pi_0(u) \leftarrow \emptyset$ for all $u \in V(G_0)$.
4: **for** $u \in V(G_0)$ **do**
5: $\quad \mathcal{H}_u \leftarrow \{H \in \mathcal{H} : u \text{ is incident to a copy of } H \text{ in } G_0 \text{ and } H \text{ appears in } G_1\}$.
6: $\quad$ **if** $|\mathcal{H}_u| \geq \frac{1}{2}|\mathcal{H}| \cdot v \cdot n^{v-1}(p\gamma^2)^e$ **then**
7: $\quad\quad$ Pick $H \leftarrow \mathcal{H}_u$ at random.
8: $\quad\quad$ $w \leftarrow$ the corresponding vertex of $u$ in the copy of $H$ in $G_1$.
9: $\quad\quad$ **if** $\neg \exists u' \neq u$ such that $\pi_0(u') = w$ **then** $\pi_0(u) \leftarrow w$. **end if**
10: $\quad$ **end if**
11: **end for**
12: **return** $\pi_0$.

**Step 3: Boosting** Finally, in the last step of the recovery algorithm, we boost the partial assignment to a full permutation from $V(G_0)$ to $V(G_1)$. This step is based on the "Percolation Graph Matching" used in works such as [12, 13, 32, 33, 14]. However, we need a stronger analysis of this step, since the partial knowledge obtained from PARTIALASSIGNMENT can be noisy and (more importantly) might have arbitrary correlation with the random graph, and hence we need to assume that it might be *adversarially chosen*. Specifically, we show that we can boost an $(\frac{n}{O(\log \log n)}, 1 - o(1))$ partial assignment to a the full ground truth:

**Lemma 4.2** (Boosting from partial knowledge). *Let $p, \gamma, n, \eta, c, \theta$ be such that $p\gamma n \geq \log^c n$ for $c > 1$, $\eta\theta = o(\gamma^2)$ and $\theta = \Omega(\log^{1-c} n)$. Then with probability $1 - o(1)$ over the choice of $(G_0, G_1, \pi^*)$ from $\mathcal{D}_{\text{struct}}(n, p; \gamma)$, if BOOSTING is given $G_0, G_1$ and any $(\theta n, 1 - \eta)$ partial assignment $\pi$, then it outputs the ground truth permutation $\pi^*$.*

**Algorithm 4** BOOSTING

**Input:** Parameters $n, p, \gamma$, graphs $G_0, G_1$ sampled from $\mathcal{D}_{\text{struct}}(n, p; \gamma)$, a partial assignment $\pi_0 \in \mathcal{S}_n$.
**Output:** A permutation $\pi \in \mathcal{S}_n$.
1: $(\theta, \eta) \leftarrow$ Lemma 4.1 and $\pi \leftarrow \pi_0$. $\qquad\qquad\qquad\qquad$ ▷ $\pi_0$ is a $(\theta, \eta)$-partial assignment.
2: $\Delta \leftarrow \lfloor \theta\gamma^2 np/100 \rfloor$.
3: **for** $u \in V(G_0), w \in V(G_1)$ **do** $N(u, w) \leftarrow |\{u' \in V(G_0) : u \sim u', \pi(u') \sim w\}|$. **end for**
4: **while** $u \in V(G_0)$ where $\pi(u) = \emptyset$ and $\exists w \in V(G_1), N(u, w) \geq \Delta$ **do** $\pi(u) \leftarrow w$. **end while**
5: **if** $\pi$ is not a permutation **then** Complete $\pi$ arbitrarily. **end if**
6: $\Delta' \leftarrow \lfloor \gamma^2 np/100 \rfloor$.
7: **while** $\exists u \in V(G_0), w \in V(G_1)$ such that $N(u, w) \geq \Delta'$ and $N(u, \pi(u)), N(\pi^{-1}(w), w) < \Delta'/10$ **do** Modify $\pi$ by mapping $u$ to $w$ and mapping $\pi^{-1}(w)$ to $\pi(u)$. **end while**
8: **return** $\pi$.

## Footnotes

[2]See the surveys [9, 10], the latter of which is titled "Thirty Years of Graph Matching in Pattern Recognition".

[3]Hamiltonian path is NP hard and can be reduced to graph matching by matching the input with a cycle.

[4]Some works also studied a more general variant where $G_0$ and $G_1$ use different subsampling parameters $\gamma_0, \gamma_1$. Our work extends to this setting as well but for simplicity we focus on the $\gamma_0 = \gamma_1$ case.

[5]More formally, $v_j^b = X_{H_j}(G_b)$ where $X_H(G)$ is the number of *injective homomorphisms* of $H$ to $G$, divided by the number of *automorphisms* of $H$.

[6]The range of values of $p$ our algorithm is proven to succeed for corresponds to the degrees achievable in Theorem 2.2. We conjecture that a family achieving these properties can be obtained with any density $e/v > 1$, which would extend our analysis to $p = n^{1-\delta}$ for all $\delta \in (0, 1)$ (see the supplementary materials).

[7]This condition is a strengthening of the "strict balance" condition in the random graph literature [31].

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
