[Supplementary Material]

# (Nearly) Efficient Algorithms for the Graph Matching Problem on Correlated Random Graphs

Boaz Barak, Chi-Ning Chou, Zhixian Lei, Tselil Schramm, Yueqi Sheng
Harvard University, MA, USA

## ❑ Motivation

➤ De-anonymization (e.g., matching social networks)

Matching the users

➤ Malware detection (e.g., finding suspicious patterns in a code)

🙂 Malware-free

😱 Containing malware

## ❑ Problem Formulation

➤ The distance between two graphs: $\min_{\pi \in \mathcal{S}_n} \|G_0 - \pi(G_1)\|_0$ .

➤ Input model: **Correlated Erdös-Rényi Graphs**.

Erdös-Rényi graph

Remove each edge i.i.d. with prob. q

$G_0$

$G_1$

➤ Two computational problems:
- **Graph similarity: hypothesis testing.** Given $(G_0, G_1)$, distinguish (i) correlated Erdös-Rényi and (ii) independent Erdös-Rényi.
- **Graph matching: recovery.** Given $(G_0, G_1)$ sampled from correlated Erdös-Rényi, find the $\pi^*$ that minimizes the distance.

## ❑ Prior Work

➤ Only exponential time algorithms were known, *e.g., percolation*.

## ❑ Our Results

➤ **Graph similarity:** We give the *first polynomial time* algorithm.

➤ **Graph matching:** We give the *first quasi-polynomial time* algorithm.

| Paper | Algorithm | Runtime |
|---|---|---|
| Cullina & Kivayash | Info-theoretic | $\exp(O(n))$ |
| Yartseva & Grossglauser | percolation | $\exp((1-\delta)n)$ |
| **This work** | Subgraph matching | $n^{O(\log n)}$ * |
| Mossel & Xu | Seeded local statistics | $n^{O(\log n)}$ * |

\* The runtime does not work for all regimes. Ask me for more details!

## ❑ Our "Black Swan" Approach

➤ **Intuition:** Use a family of small graphs (a *flock of black swans*) as the features to compare $(G_0, G_1)$.

| A Swan | A Black Swan |
|---|---|
|  |  |
| The variance of #appearance is **large**. | ✓ #appearance **concentrates** near exp. |
| **Too many** automorphisms. | ✓ **Unique** automorphism. |
| **Large** overlap with other swans. | ✓ **Small** overlap with other black swans. |

➤ **Difficulties:** Construct a large family of black swans with the desiring properties.

## ❑ Algorithms

➤ **Graph similarity:** Use the *correlation of the black swan counts* to perform hypothesis testing.
- Let $\mathcal{H}$ be a family of black swans and $X_H(G)$ be the # of $H$'s in $G$.
- Define the correlation polynomial:

$$P_{\mathcal{H}}(G_0, G_1) = \frac{1}{|\mathcal{H}|} \sum_{H \in \mathcal{H}} (X_H(G_0) - \mathbb{E}_G X_H(G))(X_H(G_1) - \mathbb{E}_G X_H(G)).$$

- **(Correlated Erdös-Rényi):** $|P_{\mathcal{H}}(G_0, G_1)|$ is large.
- **(Independent Correlated Erdös-Rényi):** $|P_{\mathcal{H}}(G_0, G_1)|$ is small.

➤ **Graph matching:** For each vertex v, the black swan family gives a *signature vector* according to the position of v in each swan.
- **(Partial assignment):** The uniqueness of each swan guarantees the signature vector from $G_0$ and $G_1$ of the same vertex being close. This holds w.h.p. for many vertices and give a partial assignment.
- **(Boosting):** Use the partial assignment as the seeds and generate a full permutation that matches $G_0$ and $G_1$.

## ❑ Future Directions

➤ **For theorists:** (i) Improve the runtime, (ii) construct black swans for a larger range of parameters, and (iii) computational limitation.

➤ **For experimentalists:** Can our black swan approach guide practical algorithms for graph matching?

- - - - - - - - - - - - - - - - - - - - - - - - - - - - - - - - - - - - - - - - -

- Conference version:
- arXiv version: