[Reviews · NeurIPS 2019]

Reviewer 1



I find the problem to be reasonable well motivated and the work non trivial. Analyzing subgraph counts is usually difficult and this work is no exception. The construction of the family of subgraph is novel and may find applications elsewhere. The paper is well written and the authors do a good job in communicating their ideas in a coherent and understandable fashion. My biggest concern is the disconnect between the theory and experiments. The theory uses a complicated list of motifs whereas the experiments use a set of five subgraphs. The theory uses random graphs whereas one experiment consider graphs from facebook which are very distinct from random graphs in many aspects (power law degree distribution, clustering and so on). I find that the assertion "We believe our techniques are of independent interest and applicability beyond the correlated ER model" to be not convincing with weak evidence. IMHO more work is needed to justify this assertion. It is not clear (to me) whether the hypothesis testing problem has been proposed before. The authors should clarify this in the text. Also for finding the permutation minimizing the error, it is not clear to me that the problem with base graph chosen in a worse-case manner (as opposed to a random graph) is NP-hard. It would be good to provide a proof (or mention this is not known). Some more comments: L4: vision--> computer vision Equation 1: I was not sure what || ||_0 means. I would specify this. L 29: In this paper--> consider deleting L 43-50. The authors repeat the applications of the problems several times in several different places. I would remove some of these. L 54: related works--> related work L 70: By no means easy to establish: I would consider rephrasing. L 72: But have not been able to so. This is a rather limited evidence for the difficulty of the problem. I would remove it. L 78-81 As I said the suggestion of relevance documented by the experimental evidence is weak at best. I think more work is needed to justify these claims. L 82-83: very much: delete L 98: Simple percolation is not very informative. I would add more details. L 155: , this can be used: Sentence needs revision. Beginning of section 2 and elsewhere. If the statement holds for all epsilon then does it hold with probability $1-o(1)$? Theorem 3.1: Specify that $\delta$ is in $(0,1)$. Also does the result hold for $\gamma<<1$ (with worst running time)? Figure 3: text is very small making it very cryptic to read. It would be helpful to have information about the running time of the various algorithms. Different benchmark algorithms where chosen for different simulations (NN for one but not the others). Justification of this or using the same algorithms throughout is recommended. References has multiple typos: It is not sorted alphabetically. [13] should have Erdos Renyi capitalized. [33] g_{n,p} need to be corrected.

Reviewer 2



This paper studies a matching problem on correlated random graphs. In this model, a Erdos-Renyi graph $Gt$ is first sampled. Then two sugbraphs are sampled from $G$ (with vertex labels shuffled). This paper considers two problem: (i) hypothesis testing: in H_0, two independent Erdos-Renyi graphs are given; in H_1, two graphs are sampled from the correlated random graph model. The goal is to design a randomized algorithm to determine whether we are in H_0 and H_1. (ii) matching: given two graphs sampled from the correlated random graph model, design an algorithm to match the nodes from two subgraphs. I find the problem well motivated. First, this problem is related to the graph isomorphism problem, which is notoriously difficult in the worst case. Thus, it is natural and important to study "average case" behaviors (i.e., design algorithms under random graph models). Second, the authors mentioned a number of settings (i.e., de-anonymize social graphs), in which the new algorithmic techniques developed in the papers can be potentially applicable. The new algorithmic techniques developed by the authors are also very interesting. As mentioned by the authors, "simple statistics" (e.g., degree distribution, eigenvalues/eigenvectors) of a random graph is usually highly concentrated so they are unsuitable for hypothesis testing. The authors propose that they can design a family of subgraphs so that these subgraphs are unlikely to be encountered in a random graph (and even less likely to appear in two independent random graphs). Then by counting the numbers of these subgraphs in the input graphs, we can determine whether H_0 needs to be rejected. The technique can be generalized to design a matching algorithm. Designing the family of subgraphs for counting seems to be highly non-trivial for a number of reasons: (i) the graphs need to be "unnatural" so that they appear with low probability; indeed, the authors find an interesting way to build paths on top of random graphs to make them "unnatural"; and (ii) the size of these subgraphs need to be a constant (to make sure the running time is polynomial). It is usually harder to manipulate constant size random graphs to get "high probability" results. In summary, I find this paper studies a very interesting theoretical model and proposes a new algorithmic technique that can potentially have practical impact. It is a typical theory paper in first tier ML conferences such as Neurips/COLT.

Reviewer 3



This is an interesting paper on a timely topic. As far as I understood, the results are valid in the dens case (i.e. average degree of the order n^\delta) and for fixed values of \gamma. It seems that recent works like: Efficient random graph matching via degree profiles by Ding et al (https://arxiv.org/abs/1811.07821) allows to get better theoretical results. -------------- Post feedback: I've read the feedback of the authors. I think that the main contribution of the paper is theory and the experiment section could be removed.

[Author Response · NeurIPS 2019]

First, thanks to the reviewers for the thorough and thoughtful reviews and comments!

**Experiments:** We stress (as we have tried to do in the paper) that our main contribution is theoretical. Our experiments are very preliminary. On the other hand, we were pleasantly surprised to find that our techniques, which we developed for the Erdös-Rényi (ER) model, have performance comparable to other algorithms (which have no theoretical guarantees) on real data. We include the experiments because they suggest that getting practical algorithms based on subgraph counts for real instances is a tenable (and in our opinion, interesting) direction for future research.

**Responses to reviewer 1:** The reviewer said: "the assertion "We believe our techniques are of independent interest and applicability beyond the correlated ER model" [...] IMHO more work is needed to justify this assertion." We agree that more work is needed to pronounce that subgraph counting algorithms are successful beyond ER graphs; we did not intend to suggest otherwise! See "Experiments" above.

Regarding the relation to prior literature:

- For worst-case graphs the matching problem is NP-hard; thanks for pointing out the omission. One can reduce from Hamiltonian cycle, taking $G_0$ to be the target graph and taking $G_1$ to be a cycle on $n$ vertices.
- The worst-case hypothesis testing problem was previously studied in the context of malware detection (see e.g. Section 2.2 of [Park-Reeves-Mulukutla-Sundaravel'10]). We are unaware of prior work on this question for random graphs.
- We will add a comparison between our work and that of Applebaum et al. and also Bhaskara et al. These papers do face some similar technical challenges having to do with subgraph counts in random graphs, but our goal is sufficiently different that it seems their techniques do not apply (at least not more naturally than what we already do).
- We should have mentioned the average-case graph isomorphism paper of Babai. The techniques there don't apply since the degree sequence (and in general neighborhood structure) cannot tolerate more than $o(1)$ noise.

The reviewer suggests that we "show that using simper family of black swans (cliques, cycles) does NOT work." We can show this in the relevant correlation regime (correlation $\gamma < 1 - \epsilon$). This because for any one subgraph $H$ on $v$ vertices, the variance in the subgraph count overwhelms the correlation by a multiplicative factor of $(1/\gamma)^{O(v)}$. To decrease the variance, we must take a family of size exponential in $v$ (more precisely $(1/\gamma)^{O(v)}$). Since there are only $O(v)$ cycles and cliques on up to $v$ vertices, there are not enough such "simple" graphs to reduce the variance.

**Response to reviewer 2:** No concrete questions were posed. Thanks for the review!

**Response to reviewer 3:** The recent work "Efficient random graph matching via degree profiles" by Ding et al. is *not* comparable with this work. Their algorithm tolerates significantly less noise: they require that two graphs differ on at most a $O(\frac{1}{\log^2 n})$ fraction of edges where $n$ is the size of the base graph. In our paper we can tolerate a constant fraction of mismatched edges but still achieve exact recovery.

The reviewer says that the arXiv version is better-written and has more rigorous theorem statements. We agree; due to page limits we simplified. We will make the theorem statement more formal in the final version.

To compare our algorithm with GNN and SimGNN, we recast hypothesis testing as a classification problem. In the training phase, we generate pairs of graphs from the null hypothesis and the alternative hypothesis and feed them into neural network for supervised learning. Next we test the neural networks by asking them to classify newly generated pairs of graphs from both cases. The other papers mentioned by the reviewer are indeed relevant, and we will consider and incorporate comparisons into the final version of the paper.

For the suggestion about comparing our algorithm with those of by Feizi et al. and Dai et al.: these papers are clearly related to the problems we studied. The reason that spectral algorithms or canonical labeling algorithms don't work well is due to our high noise rate. For example, for two $\frac{1}{10}$-correlated graphs $G_1$ and $G_2$, the top eigenvector of $G_1$ and $G_2$ will be $\approx \frac{1}{10}$-correlated. In canonical labeling, the performance depends on the statistics used for labeling the vertices. If we label vertices just by degree statistics, there is no significant improvement can be made over the degree sequence approach. The reason our approach handles high noise is that we aggregate many statistics to reduce the variance.

The goal of the Facebook experiment is to give evidence that the subgraph count approach may work beyond the ER model, and can be useful for identifying the matching between two similar unlabeled "real-world" graphs. It is a simple algorithm and it achieves non-trivial number of correct matchings, even though the Facebook graph is dissimilar to a random graph. Greater improvements may be possible if the approach is developed further, and we find this an interesting direction for future research.

[Meta-Review · NeurIPS 2019]

The reviewers are all positive about the paper. The authors should seriously consider whether Section 5 in the paper as it currently stands is suitable. There is a view among the reviewers that it does more harm than good. Experiments are not really necessary in a NeurIPS paper, and if the gap between the theory set-up and the experimental set-up is large, it is probably worth removing them altogether. In any case, a proper discussion should be added if the section is retained.